# Coresets for Scalable Bayesian Logistic Regression

**Jonathan H. Huggins**     **Trevor Campbell**     **Tamara Broderick**
Computer Science and Artificial Intelligence Laboratory, MIT
{jhuggins@, tdjc@, tbroderick@csail.}mit.edu

## Abstract

The use of Bayesian methods in large-scale data settings is attractive because of the rich hierarchical models, uncertainty quantification, and prior specification they provide. Standard Bayesian inference algorithms are computationally expensive, however, making their direct application to large datasets difficult or infeasible. Recent work on scaling Bayesian inference has focused on modifying the underlying algorithms to, for example, use only a random data subsample at each iteration. We leverage the insight that data is often redundant to instead obtain a weighted subset of the data (called a *coreset*) that is much smaller than the original dataset. We can then use this small coreset in any number of existing posterior inference algorithms without modification. In this paper, we develop an efficient coreset construction algorithm for Bayesian logistic regression models. We provide theoretical guarantees on the size and approximation quality of the coreset – both for fixed, known datasets, and in expectation for a wide class of data generative models. Crucially, the proposed approach also permits efficient construction of the coreset in both streaming and parallel settings, with minimal additional effort. We demonstrate the efficacy of our approach on a number of synthetic and real-world datasets, and find that, in practice, the size of the coreset is independent of the original dataset size. Furthermore, constructing the coreset takes a negligible amount of time compared to that required to run MCMC on it.

## 1   Introduction

Large-scale datasets, comprising tens or hundreds of millions of observations, are becoming the norm in scientific and commercial applications ranging from population genetics to advertising. At such scales even simple operations, such as examining each data point a small number of times, become burdensome; it is sometimes not possible to fit all data in the physical memory of a single machine. These constraints have, in the past, limited practitioners to relatively simple statistical modeling approaches. However, the rich hierarchical models, uncertainty quantification, and prior specification provided by Bayesian methods have motivated substantial recent effort in making Bayesian inference procedures, which are often computationally expensive, scale to the large-data setting.

The standard approach to Bayesian inference for large-scale data is to modify a specific inference algorithm, such as MCMC or variational Bayes, to handle distributed or streaming processing of data. Examples include subsampling and streaming methods for variational Bayes [6, 7, 16], subsampling methods for MCMC [4, 18, 24], and distributed "consensus" methods for MCMC [8, 19, 21, 22]. Existing methods, however, suffer from both practical and theoretical limitations. Stochastic variational inference [16] and subsampling MCMC methods use a new random subset of the data at each iteration, which requires random access to the data and hence is infeasible for very large datasets that do not fit into memory. Furthermore, in practice, subsampling MCMC methods have been found to require examining a constant fraction of the data at each iteration, severely limiting the computational gains obtained [5, 23]. More scalable methods such as consensus MCMC [19, 21, 22]

and streaming variational Bayes [6, 7] lead to gains in computational efficiency, but lack rigorous justification and provide no guarantees on the quality of inference.

An important insight in the large-scale setting is that much of the data is often *redundant*, though there may also be a small set of data points that are distinctive. For example, in a large document corpus, one news article about a hockey game may serve as an excellent representative of hundreds or thousands of other similar pieces about hockey games. However, there may only be a few articles about luge, so it is also important to include at least one article about luge. Similarly, one individual's genetic information may serve as a strong representative of other individuals from the same ancestral population admixture, though some individuals may be genetic outliers. We leverage data redundancy to develop a scalable Bayesian inference framework that modifies the *dataset* instead of the common practice of modifying the inference algorithm. Our method, which can be thought of as a preprocessing step, constructs a *coreset* – a small, weighted subset of the data that approximates the full dataset [1, 9] – that can be used in many standard inference procedures to provide posterior approximations with guaranteed quality. The scalability of posterior inference with a coreset thus simply depends on the coreset's growth with the full dataset size. To the best of our knowledge, coresets have not previously been used in a Bayesian setting.

The concept of coresets originated in computational geometry (e.g. [1]), but then became popular in theoretical computer science as a way to efficiently solve clustering problems such as $k$-means and PCA (see [9, 11] and references therein). Coreset research in the machine learning community has focused on scalable clustering in the optimization setting [3, 17], with the exception of Feldman et al. [10], who developed a coreset algorithm for Gaussian mixture models. Coreset-like ideas have previously been explored for maximum likelihood-learning of logistic regression models, though these methods either lack rigorous justification or have only asymptotic guarantees (see [15] and references therein).

The job of the coreset in the Bayesian setting is to provide an approximation of the full data log-likelihood up to a multiplicative error uniformly over the parameter space. As this paper is the first foray into applying coresets in Bayesian inference, we begin with a theoretical analysis of the quality of the posterior distribution obtained from such an approximate log-likelihood. The remainder of the paper develops the efficient construction of small coresets for Bayesian logistic regression, a useful and widely-used model for the ubiquitous problem of binary classification. We develop a coreset construction algorithm, the output of which uniformly approximates the full data log-likelihood over parameter values in a ball with a user-specified radius. The approximation guarantee holds for a given dataset with high probability. We also obtain results showing that the boundedness of the parameter space is necessary for the construction of a nontrivial coreset, as well as results characterizing the algorithm's expected performance under a wide class of data-generating distributions. Our proposed algorithm is applicable in both the streaming and distributed computation settings, and the coreset can then be used by any inference algorithm which accesses the (gradient of the) log-likelihood as a black box. Although our coreset algorithm is specifically for logistic regression, our approach is broadly applicable to other Bayesian generative models.

Experiments on a variety of synthetic and real-world datasets validate our approach and demonstrate robustness to the choice of algorithm hyperparameters. An empirical comparison to random subsampling shows that, in many cases, coreset-based posteriors are orders of magnitude better in terms of maximum mean discrepancy, including on a challenging 100-dimensional real-world dataset. Crucially, our coreset construction algorithm adds negligible computational overhead to the inference procedure. All proofs are deferred to the Supplementary Material.

## 2 Problem Setting

We begin with the general problem of Bayesian posterior inference. Let $\mathcal{D} = \{(X_n, Y_n)\}_{n=1}^N$ be a dataset, where $X_n \in \mathcal{X}$ is a vector of covariates and $Y_n \in \mathcal{Y}$ is an observation. Let $\pi_0(\theta)$ be a prior density on a parameter $\theta \in \Theta$ and let $p(Y_n \mid X_n, \theta)$ be the likelihood of observation $n$ given the parameter $\theta$. The Bayesian posterior is given by the density $\pi_N(\theta)$, where

$$\pi_N(\theta) := \frac{\exp(\mathcal{L}_N(\theta))\pi_0(\theta)}{\mathcal{E}_N}, \quad \mathcal{L}_N(\theta) := \sum_{n=1}^N \ln p(Y_n \mid X_n, \theta), \quad \mathcal{E}_N := \int \exp(\mathcal{L}_N(\theta))\pi_0(\theta)\, \mathrm{d}\theta.$$

---
**Algorithm 1** Construction of logistic regression coreset
---
**Require:** Data $\mathcal{D}$, $k$-clustering $\mathcal{Q}$, radius $R > 0$, tolerance $\varepsilon > 0$, failure rate $\delta \in (0, 1)$

1: **for** $n = 1, \ldots, N$ **do**  $\qquad\qquad\qquad$ ▷ calculate sensitivity upper bounds using the $k$-clustering
2: $\quad m_n \leftarrow \dfrac{N}{1 + \sum_{i=1}^{k} |G_i^{(-n)}| e^{-R\|\bar{Z}_{G,i}^{(-n)} - Z_n\|_2}}$
3: **end for**
4: $\bar{m}_N \leftarrow \frac{1}{N} \sum_{n=1}^{N} m_n$
5: $M \leftarrow \left\lceil \frac{c\bar{m}_N}{\varepsilon^2} [(D+1) \log \bar{m}_N + \log(1/\delta)] \right\rceil$  $\qquad$ ▷ coreset size; $c$ is from proof of Theorem B.1
6: **for** $n = 1, \ldots, N$ **do**
7: $\quad p_n \leftarrow \frac{m_n}{N\bar{m}_N}$  $\qquad\qquad\qquad\qquad\qquad\qquad\qquad\qquad$ ▷ importance weights of data
8: **end for**
9: $(K_1, \ldots, K_N) \sim \mathsf{Multi}(M, (p_n)_{n=1}^{N})$  $\qquad\qquad\qquad\qquad$ ▷ sample data for coreset
10: **for** $n = 1, \ldots, N$ **do**  $\qquad\qquad\qquad\qquad\qquad\qquad\qquad$ ▷ calculate coreset weights
11: $\quad \gamma_n \leftarrow \frac{K_n}{p_n M}$
12: **end for**
13: $\tilde{\mathcal{D}} \leftarrow \{(\gamma_n, X_n, Y_n) \,|\, \gamma_n > 0\}$  $\qquad\qquad\qquad$ ▷ only keep data points with non-zero weights
14: **return** $\tilde{\mathcal{D}}$
---

Our aim is to construct a weighted dataset $\tilde{\mathcal{D}} = \{(\gamma_m, \tilde{X}_m, \tilde{Y}_m)\}_{m=1}^{M}$ with $M \ll N$ such that the weighted log-likelihood $\tilde{\mathcal{L}}_N(\theta) = \sum_{m=1}^{M} \gamma_m \ln p(\tilde{Y}_n \,|\, \tilde{X}_m, \theta)$ satisfies

$$|\mathcal{L}_N(\theta) - \tilde{\mathcal{L}}_N(\theta)| \leq \varepsilon |\mathcal{L}_N(\theta)|, \quad \forall \theta \in \Theta. \tag{1}$$

If $\tilde{\mathcal{D}}$ satisfies Eq. (1), it is called an $\varepsilon$-*coreset of* $\mathcal{D}$, and the approximate posterior

$$\tilde{\pi}_N(\theta) = \frac{\exp(\tilde{\mathcal{L}}_N(\theta))\pi_0(\theta)}{\tilde{\mathcal{E}}_N}, \qquad \text{with} \quad \tilde{\mathcal{E}}_N = \int \exp(\tilde{\mathcal{L}}_N(\theta))\pi_0(\theta)\,\mathrm{d}\theta,$$

has a marginal likelihood $\tilde{\mathcal{E}}_N$ which approximates the true marginal likelihood $\mathcal{E}_N$, shown by Proposition 2.1. Thus, from a Bayesian perspective, the $\varepsilon$-coreset is a useful notion of approximation.

**Proposition 2.1.** *Let $\mathcal{L}(\theta)$ and $\tilde{\mathcal{L}}(\theta)$ be arbitrary non-positive log-likelihood functions that satisfy $|\mathcal{L}(\theta) - \tilde{\mathcal{L}}(\theta)| \leq \varepsilon |\mathcal{L}(\theta)|$ for all $\theta \in \Theta$. Then for any prior $\pi_0(\theta)$ such that the marginal likelihoods*

$$\mathcal{E} = \int \exp(\mathcal{L}(\theta))\pi_0(\theta)\,\mathrm{d}\theta \qquad \text{and} \qquad \tilde{\mathcal{E}} = \int \exp(\tilde{\mathcal{L}}(\theta))\pi_0(\theta)\,\mathrm{d}\theta$$

*are finite, the marginal likelihoods satisfy* $|\ln \mathcal{E} - \ln \tilde{\mathcal{E}}| \leq \varepsilon |\ln \mathcal{E}|$.

## 3  Coresets for Logistic Regression

### 3.1  Coreset Construction

In logistic regression, the covariates are real feature vectors $X_n \in \mathbb{R}^D$, the observations are labels $Y_n \in \{-1, 1\}$, $\Theta \subseteq \mathbb{R}^D$, and the likelihood is defined as

$$p(Y_n \,|\, X_n, \theta) = p_{logistic}(Y_n \,|\, X_n, \theta) := \frac{1}{1 + \exp(-Y_n X_n \cdot \theta)}.$$

The analysis in this work allows any prior $\pi_0(\theta)$; common choices are the Gaussian, Cauchy [12], and spike-and-slab [13]. For notational brevity, we define $Z_n := Y_n X_n$, and let $\phi(s) := \ln(1 + \exp(-s))$. Choosing the optimal $\epsilon$-coreset is not computationally feasible, so we take a less direct approach. We design our coreset construction algorithm and prove its correctness using a quantity $\sigma_n(\Theta)$ called the *sensitivity* [9], which quantifies the redundancy of a particular data point $n$ – the larger the sensitivity, the less redundant. In the setting of logistic regression, we have that the sensitivity is

$$\sigma_n(\Theta) := \sup_{\theta \in \Theta} \frac{N \, \phi(Z_n \cdot \theta)}{\sum_{\ell=1}^{N} \phi(Z_\ell \cdot \theta)}.$$

Intuitively, $\sigma_n(\Theta)$ captures how much influence data point $n$ has on the log-likelihood $\mathcal{L}_N(\theta)$ when varying the parameter $\theta \in \Theta$, and thus data points with high sensitivity should be included in the coreset. Evaluating $\sigma_n(\Theta)$ exactly is not tractable, however, so an upper bound $m_n \geq \sigma_n(\Theta)$ must be used in its place. Thus, the key challenge is to efficiently compute a tight upper bound on the sensitivity.

For the moment we will consider $\Theta = \mathbb{B}_R$ for any $R > 0$, where $\mathbb{B}_R := \{\theta \in \mathbb{R}^D \mid \|\theta\|_2 \leq R\}$; We discuss the case of $\Theta = \mathbb{R}^D$ shortly. Choosing the parameter space to be a Euclidean ball is reasonable since data is usually preprocessed to have mean zero and variance 1 (or, for sparse data, to be between -1 and 1), so each component of $\theta$ is typically in a range close to zero (e.g. between -4 and 4) [12].

The idea behind our sensitivity upper bound construction is that we would expect data points that are bunched together to be redundant while data points that are far from from other data have a large effect on inferences. Clustering is an effective way to summarize data and detect outliers, so we will use a *k-clustering* of the data $\mathcal{D}$ to construct the sensitivity bound. A $k$-clustering is given by $k$ cluster centers $\mathcal{Q} = \{Q_1, \ldots, Q_k\}$. Let $G_i := \{Z_n \mid i = \arg\min_j \|Q_j - Z_n\|_2\}$ be the set of vectors closest to center $Q_i$ and let $G_i^{(-n)} := G_i \setminus \{Z_n\}$. Define $Z_{G,i}^{(-n)}$ to be a uniform random vector from $G_i^{(-n)}$ and let $\bar{Z}_{G,i}^{(-n)} := \mathbb{E}[Z_{G,i}^{(-n)}]$ be its mean. The following lemma uses a $k$-clustering to establish an efficiently computable upper bound on $\sigma_n(\mathbb{B}_R)$:

**Lemma 3.1.** *For any $k$-clustering $\mathcal{Q}$,*

$$\sigma_n(\mathbb{B}_R) \leq m_n := \frac{N}{1 + \sum_{i=1}^{k} |G_i^{(-n)}| e^{-R\|\bar{Z}_{G,i}^{(-n)} - Z_n\|_2}}. \tag{2}$$

*Furthermore, $m_n$ can be calculated in $O(k)$ time.*

The bound in Eq. (2) captures the intuition that if the data forms tight clusters (that is, each $Z_n$ is close to one of the cluster centers), we expect each cluster to be well-represented by a small number of typical data points. For example, if $Z_n \in G_i$, $\|\bar{Z}_{G,i}^{(-n)} - Z_n\|_2$ is small, and $|G_i^{(-n)}| = \Theta(N)$, then $\sigma_n(\mathbb{B}_R) = O(1)$. We use the (normalized) sensitivity bounds obtained from Lemma 3.1 to form an importance distribution $(p_n)_{n=1}^{N}$ from which to sample the coreset. If we sample $Z_n$, then we assign it weight $\gamma_n$ proportional to $1/p_n$. The size of the coreset depends on the mean sensitivity bound, the desired error $\varepsilon$, and a quantity closely related to the VC dimension of $\theta \mapsto \phi(\theta \cdot Z)$, which we show is $D + 1$. Combining these pieces we obtain Algorithm 1, which constructs an $\varepsilon$-coreset with high probability by Theorem 3.2.

**Theorem 3.2.** *Fix $\varepsilon > 0$, $\delta \in (0, 1)$, and $R > 0$. Consider a dataset $\mathcal{D}$ with $k$-clustering $\mathcal{Q}$. With probability at least $1 - \delta$, Algorithm 1 with inputs $(\mathcal{D}, \mathcal{Q}, R, \varepsilon, \delta)$ constructs an $\varepsilon$-coreset of $\mathcal{D}$ for logistic regression with parameter space $\Theta = \mathbb{B}_R$. Furthermore, Algorithm 1 runs in $O(Nk)$ time.*

*Remark* 3.3. The coreset algorithm is efficient with an $O(Nk)$ running time. However, the algorithm requires a $k$-clustering, which must also be constructed. A high-quality clustering can be obtained cheaply via $k$-means++ in $O(Nk)$ time [2], although a coreset algorithm could also be used.

Examining Algorithm 1, we see that the coreset size $M$ is of order $\bar{m}_N \log \bar{m}_N$, where $\bar{m}_N = \frac{1}{N} \sum_n m_n$. So for $M$ to be smaller than $N$, at a minimum, $\bar{m}_N$ should satisfy $\bar{m}_N = \tilde{o}(N)$,[1] and preferably $\bar{m}_N = O(1)$. Indeed, for the coreset size to be small, it is critical that (a) $\Theta$ is chosen such that most of the sensitivities satisfy $\sigma_n(\Theta) \ll N$ (since $N$ is the maximum possible sensitivity), (b) each upper bound $m_n$ is close to $\sigma_n(\Theta)$, and (c) ideally, that $\bar{m}_N$ is bounded by a constant. In Section 3.2, we address (a) by providing sensitivity lower bounds, thereby showing that the constraint $\Theta = \mathbb{B}_R$ is necessary for nontrivial sensitivities even for "typical" (i.e. non-pathological) data. We then apply our lower bounds to address (b) and show that our bound in Lemma 3.1 is nearly tight. In Section 3.3, we address (c) by establishing the expected performance of the bound in Lemma 3.1 for a wide class of data-generating distributions.

## 3.2 Sensitivity Lower Bounds

We now develop lower bounds on the sensitivity to demonstrate that essentially we must limit ourselves to bounded $\Theta$,[2] thus making our choice of $\Theta = \mathbb{B}_R$ a natural one, and to show that the sensitivity upper bound from Lemma 3.1 is nearly tight.

We begin by showing that in both the worst case and the average case, for all $n$, $\sigma_n(\mathbb{R}^D) = N$, the maximum possible sensitivity – even when the $Z_n$ are arbitrarily close. Intuitively, the reason for the worst-case behavior is that if there is a separating hyperplane between a data point $Z_n$ and the remaining data points, and $\theta$ is in the direction of that hyperplane, then when $\|\theta\|_2$ becomes very large, $Z_n$ becomes arbitrarily more important than any other data point.

**Theorem 3.4.** *For any $D \geq 3$, $N \in \mathbb{N}$ and $0 < \epsilon' < 1$, there exists $\epsilon > 0$ and unit vectors $Z_1, \ldots, Z_N \in \mathbb{R}^D$ such that for all pairs $n, n'$, $Z_n \cdot Z_{n'} \geq 1 - \epsilon'$ and for all $R > 0$ and $n$,*

$$\sigma_n(\mathbb{B}_R) \geq \frac{N}{1 + (N-1)e^{-R\epsilon\sqrt{\epsilon'}/4}}, \qquad \text{and hence} \qquad \sigma_n(\mathbb{R}^D) = N.$$

The proof of Theorem 3.4 is based on choosing $N$ distinct unit vectors $V_1, \ldots, V_N \in \mathbb{R}^{D-1}$ and setting $\epsilon = 1 - \max_{n \neq n'} V_n \cdot V_{n'} > 0$. But what is a "typical" value for $\epsilon$? In the case of the vectors being uniformly distributed on the unit sphere, we have the following scaling for $\epsilon$ as $N$ increases:

**Proposition 3.5.** *If $V_1, \ldots, V_N$ are independent and uniformly distributed on the unit sphere $\mathbb{S}^D := \{v \in \mathbb{R}^D \mid \|v\| = 1\}$ with $D \geq 2$, then with high probability*

$$1 - \max_{n \neq n'} V_n \cdot V_{n'} \geq C_D N^{-4/(D-1)},$$

*where $C_D$ is a constant depending only on $D$.*

Furthermore, $N$ can be exponential in $D$ even with $\epsilon$ remaining very close to 1:

**Proposition 3.6.** *For $N = \lfloor \exp((1-\epsilon)^2 D/4)/\sqrt{2} \rfloor$, and $V_1, \ldots, V_N$ i.i.d. such that $V_{ni} = \pm \frac{1}{\sqrt{D}}$ with probability $1/2$, then with probability at least $1/2$, $1 - \max_{n \neq n'} V_n \cdot V_{n'} \geq \epsilon$.*

Propositions 3.5 and 3.6 demonstrate that the data vectors $Z_n$ found in Theorem 3.4 are, in two different senses, "typical" vectors and should not be thought of as worst-case data only occurring in some "negligible" or zero-measure set. These three results thus demonstrate that it is necessary to restrict attention to bounded $\Theta$. We can also use Theorem 3.4 to show that our sensitivity upper bound is nearly tight.

**Corollary 3.7.** *For the data $Z_1, \ldots, Z_N$ from Theorem 3.4,*

$$\frac{N}{1 + (N-1)e^{-R\epsilon\sqrt{\epsilon'}/4}} \leq \sigma_n(\mathbb{B}_R) \leq \frac{N}{1 + (N-1)e^{-R\sqrt{2\epsilon'}}}.$$

## 3.3 $k$-Clustering Sensitivity Bound Performance

While Lemma 3.1 and Corollary 3.7 provide an upper bound on the sensitivity given a fixed dataset, we would also like to understand how the expected mean sensitivity increases with $N$. We might expect it to be finite since the logistic regression likelihood model is parametric; the coreset would thus be acting as a sort of approximate finite sufficient statistic. Proposition 3.8 characterizes the expected performance of the upper bound from Lemma 3.1 under a wide class of generating distributions. This result demonstrates that, under reasonable conditions, the expected value of $\bar{m}_N$ is bounded for all $N$. As a concrete example, Corollary 3.9 specializes Proposition 3.8 to data with a single shared Gaussian generating distribution.

**Proposition 3.8.** *Let $X_n \overset{indep}{\sim} \mathcal{N}(\mu_{L_n}, \Sigma_{L_n})$, where $L_n \overset{indep}{\sim} \mathrm{Multi}(\pi_1, \pi_2, \ldots)$ is the mixture component responsible for generating $X_n$. For $n = 1, \ldots, N$, let $Y_n \in \{-1, 1\}$ be conditionally independent given $X_n$ and set $Z_n = Y_n X_n$. Select $0 < r < 1/2$, and define $\eta_i = \max(\pi_i - N^{-r}, 0)$. The clustering of the data implied by $(L_n)_{n=1}^N$ results in the expected sensitivity bound*

$$\mathbb{E}[\bar{m}_N] \leq \frac{1}{N^{-1} + \sum_i \eta_i e^{-R\sqrt{A_i N^{-1} \eta_i^{-1} + B_i}}} + \sum_{i:\eta_i > 0} N e^{-2N^{1-2r}} \overset{N \to \infty}{\to} \frac{1}{\sum_i \pi_i e^{-R\sqrt{B_i}}},$$

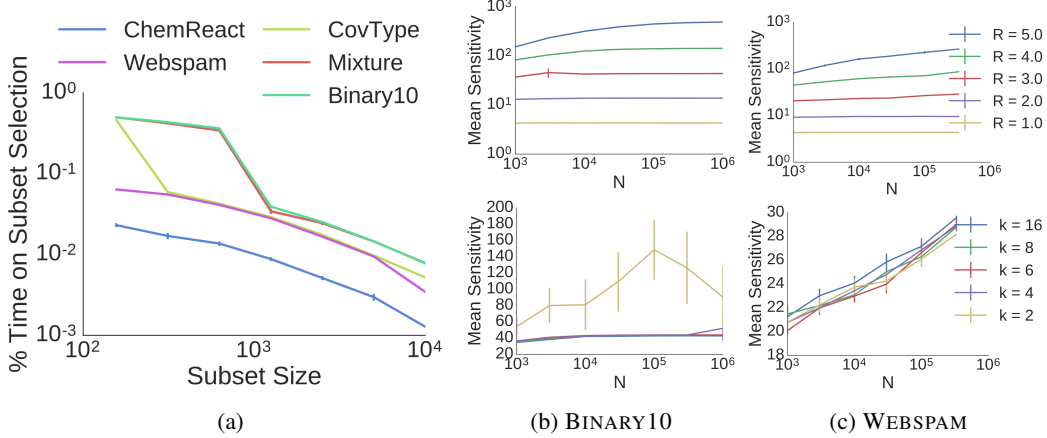

(a)                              (b) BINARY10              (c) WEBSPAM

Figure 1: **(a)** Percentage of time spent creating the coreset relative to the total inference time (including 10,000 iterations of MCMC). Except for very small coreset sizes, coreset construction is a small fraction of the overall time. **(b,c)** The mean sensitivities for varying choices of $R$ and $k$. When $R$ varies $k = 6$ and when $k$ varies $R = 3$. The mean sensitivity increases exponentially in $R$, as expected, but is robust to the choice of $k$.

where $A_i := \text{Tr}[\Sigma_i] + (1 - \bar{y}_i^2)\mu_i^T\mu_i$, $B_i := \sum_j \pi_j (\text{Tr}[\Sigma_j] + \bar{y}_j^2\mu_i^T\mu_i - 2\bar{y}_i\bar{y}_j\mu_i^T\mu_j + \mu_j^T\mu_j)$, and $\bar{y}_j = \mathbb{E}[Y_1|L_1 = j]$.

**Corollary 3.9.** *In the setting of Proposition 3.8, if $\pi_1 = 1$ and all data is assigned to a single cluster, then there is a constant $C$ such that for sufficiently large $N$, $\mathbb{E}[\bar{m}_N] \leq Ce^{R\sqrt{\text{Tr}[\Sigma_1]+(1-\bar{y}_1^2)\mu_1^T\mu_1}}$.*

### 3.4   Streaming and Parallel Settings

Algorithm 1 is a batch algorithm, but it can easily be used in parallel and streaming computation settings using standard methods from the coreset literature, which are based on the following two observations (cf. [10, Section 3.2]):

1. If $\tilde{\mathcal{D}}_i$ is an $\varepsilon$-coreset for $\mathcal{D}_i$, $i = 1, 2$, then $\tilde{\mathcal{D}}_1 \cup \tilde{\mathcal{D}}_2$ is an $\varepsilon$-coreset for $\mathcal{D}_1 \cup \mathcal{D}_2$.

2. If $\tilde{\mathcal{D}}$ is an $\varepsilon$-coreset for $\mathcal{D}$ and $\tilde{\mathcal{D}}'$ is an $\varepsilon'$-coreset for $\tilde{\mathcal{D}}$, then $\tilde{\mathcal{D}}'$ is an $\varepsilon''$-coreset for $\mathcal{D}$, where $\varepsilon'' := (1 + \varepsilon)(1 + \varepsilon') - 1$.

We can use these observations to merge coresets that were constructed either in parallel, or sequentially, in a binary tree. Coresets are computed for two data blocks, merged using observation 1, then compressed further using observation 2. The next two data blocks have coresets computed and merged/compressed in the same manner, then the coresets from blocks 1&2 and 3&4 can be merged/compressed analogously. We continue in this way and organize the merge/compress operations into a binary tree. Then, if there are $B$ data blocks total, only $\log B$ blocks ever need be maintained simultaneously. In the streaming setting we would choose blocks of constant size, so $B = O(N)$, while in the parallel setting $B$ would be the number of machines available.

## 4   Experiments

We evaluated the performance of the logistic regression coreset algorithm on a number of synthetic and real-world datasets. We used a maximum dataset size of 1 million examples because we wanted to be able to calculate the true posterior, which would be infeasible for extremely large datasets.

**Synthetic Data.** We generated synthetic binary data according to the model $X_{nd} \overset{\text{indep}}{\sim} \text{Bern}(p_d), d = 1, \ldots, D$ and $Y_n \overset{\text{indep}}{\sim} p_{logistic}(\cdot \,|\, X_n, \theta)$. The idea is to simulate data in which there are a small number of rarely occurring but highly predictive features, which is a common real-world phenomenon. We thus took $p = (1, .2, .3, .5, .01, .1, .2, .007, .005, .001)$ and $\theta = (-3, 1.2, -.5, .8, 3, -1., -.7, 4, 3.5, 4.5)$ for the $D = 10$ experiments (BINARY10) and the first 5 components of $p$ and $\theta$ for the $D = 5$ experiments (BINARY5). The generative model is the same one used by Scott et al. [21] and the first 5 components of $p$ and $\theta$ correspond to those used in the

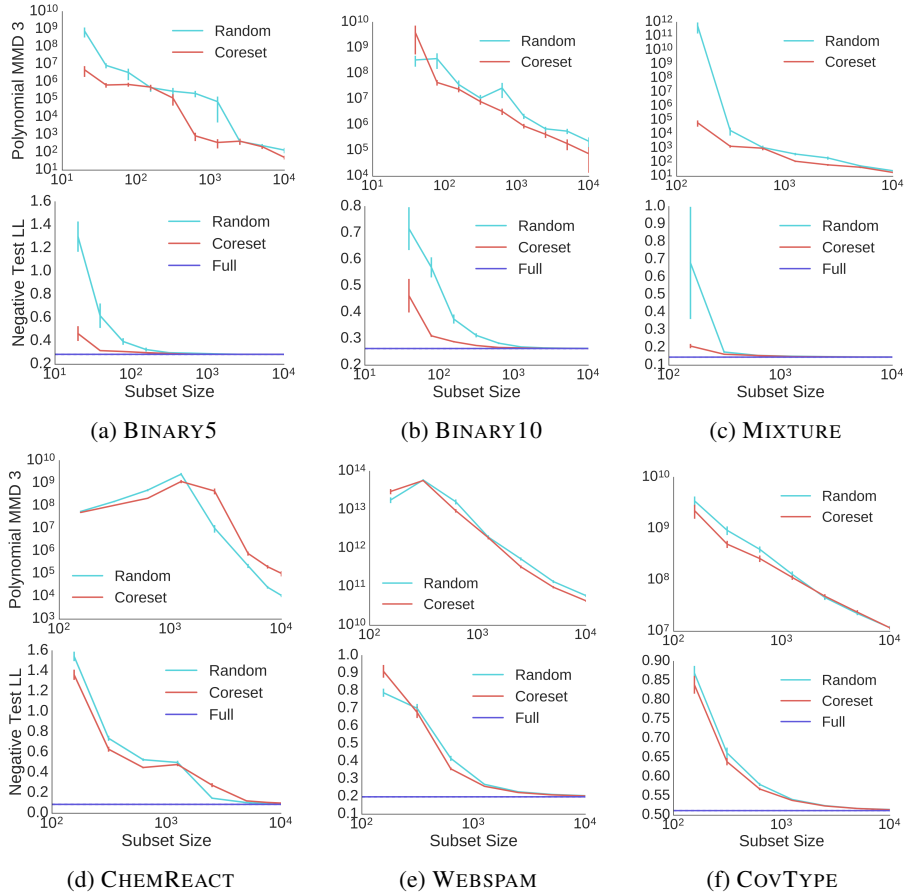

Figure 2: Polynomial MMD and negative test log-likelihood of random sampling and the logistic regression coreset algorithm for synthetic and real data with varying subset sizes (lower is better for all plots). For the synthetic data, $N = 10^6$ total data points were used and $10^3$ additional data points were generated for testing. For the real data, 2,500 (resp. 50,000 and 29,000) data points of the CHEMREACT (resp. WEBSPAM and COVTYPE) dataset were held out for testing. One standard deviation error bars were obtained by repeating each experiment 20 times.

Scott et al. experiments (given in [21, Table 1b]). We generated a synthetic mixture dataset with continuous covariates (MIXTURE) using a model similar to that of Han et al. [15]: $Y_n \overset{\text{i.i.d.}}{\sim} \text{Bern}(1/2)$ and $X_n \overset{\text{indep}}{\sim} \mathcal{N}(\mu_{Y_n}, I)$, where $\mu_{-1} = (0,0,0,0,0,1,1,1,1,1)$ and $\mu_1 = (1,1,1,1,1,0,0,0,0,0)$.

**Real-world Data.** The CHEMREACT dataset consists of $N = 26{,}733$ chemicals, each with $D = 100$ properties. The goal is to predict whether each chemical is reactive. The WEBSPAM corpus consists of $N = 350{,}000$ web pages, approximately 60% of which are spam. The covariates consist of the $D = 127$ features that each appear in at least 25 documents. The cover type (COVTYPE) dataset consists of $N = 581{,}012$ cartographic observations with $D = 54$ features. The task is to predict the type of trees that are present at each observation location.

## 4.1 Scaling Properties of the Coreset Construction Algorithm

**Constructing Coresets.** In order for coresets to be a worthwhile preprocessing step, it is critical that the time required to construct the coreset is small relative to the time needed to complete the inference procedure. We implemented the logistic regression coreset algorithm in Python.[3] In Fig. 1a, we plot the relative time to construct the coreset for each type of dataset ($k = 6$) versus the total inference time, including 10,000 iterations of the MCMC procedure described in Section 4.2. Except for very small coreset sizes, the time to run MCMC dominates.

**Sensitivity.** An important question is how the mean sensitivity $\bar{m}_N$ scales with $N$, as it determines how the size of the coreset scales with the data. Furthermore, ensuring that mean sensitivity is robust to the number of clusters $k$ is critical since needing to adjust the algorithm hyperparameters for each dataset could lead to an unacceptable increase in computational burden. We also seek to understand how the radius $R$ affects the mean sensitivity. Figs. 1b and 1c show the results of our scaling experiments on the BINARY10 and WEBSPAM data. The mean sensitivity is essentially constant across a range of dataset sizes. For both datasets the mean sensitivity is robust to the choice of $k$ and scales exponentially in $R$, as we would expect from Lemma 3.1.

## 4.2 Posterior Approximation Quality

Since the ultimate goal is to use coresets for Bayesian inference, the key empirical question is how well a posterior formed using a coreset approximates the true posterior distribution. We compared the coreset algorithm to random subsampling of data points, since that is the approach used in many existing scalable versions of variational inference and MCMC [4, 16]. Indeed, coreset-based importance sampling could be used as a drop-in replacement for the random subsampling used by these methods, though we leave the investigation of this idea for future work.

**Experimental Setup.** We used adaptive Metropolis-adjusted Langevin algorithm (MALA) [14, 20] for posterior inference. For each dataset, we ran the coreset and random subsampling algorithms 20 times for each choice of subsample size $M$. We ran adaptive MALA for 100,000 iterations on the full dataset and each subsampled dataset. The subsampled datasets were fixed for the entirety of each run, in contrast to subsampling algorithms that resample the data at each iteration. For the synthetic datasets, which are lower dimensional, we used $k = 4$ while for the real-world datasets, which are higher dimensional, we used $k = 6$. We used a heuristic to choose $R$ as large as was feasible while still obtaining moderate total sensitivity bounds. For a clustering $\mathcal{Q}$ of data $\mathcal{D}$, let $\mathcal{I} := N^{-1} \sum_{i=1}^{k} \sum_{Z \in G_i} \|Z - Q_i\|^2$ be the normalized $k$-means score. We chose $R = a/\sqrt{\mathcal{I}}$, where $a$ is a small constant. The idea is that, for $i \in [k]$ and $Z_n \in G_i$, we want $R\|\bar{Z}_{G,i}^{(-n)} - Z_n\|_2 \approx a$ on average, so the term $\exp\{-R\|\bar{Z}_{G,i}^{(-n)} - Z_n\|_2\}$ in Eq. (2) is not too small and hence $\sigma_n(\mathbb{B}_R)$ is not too large. Our experiments used $a = 3$. We obtained similar results for $4 \leq k \leq 8$ and $2.5 \leq a \leq 3.5$, indicating that the logistic regression coreset algorithm has some robustness to the choice of these hyperparameters. We used negative test log-likelihood and maximum mean discrepancy (MMD) with a 3rd degree polynomial kernel as comparison metrics (so smaller is better).

**Synthetic Data Results.** Figures 2a-2c show the results for synthetic data. In terms of test log-likelihood, coresets did as well as or outperformed random subsampling. In terms of MMD, the coreset posterior approximation typically outperformed random subsampling by 1-2 orders of magnitude and never did worse. These results suggest much can be gained by using coresets, with comparable performance to random subsampling in the worst case.

**Real-world Data Results.** Figures 2d-2f show the results for real data. Using coresets led to better performance on CHEMREACT for small subset sizes. Because the dataset was fairly small and random subsampling was done without replacement, coresets were worse for larger subset sizes. Coreset and random subsampling performance was approximately the same for WEBSPAM. On WEBSPAM and COVTYPE, coresets either outperformed or did as well as random subsampling in terms MMD and test log-likelihood on almost all subset sizes. The only exception was that random subsampling was superior on WEBSPAM for the smallest subset set. We suspect this is due to the variance introduced by the importance sampling procedure used to generate the coreset.

For both the synthetic and real-world data, in many cases we are able to obtain a high-quality logistic regression posterior approximation using a coreset that is many orders of magnitude smaller than the full dataset – sometimes just a few hundred data points. Using such a small coreset represents a substantial reduction in the memory and computational requirements of the Bayesian inference algorithm that uses the coreset for posterior inference. We expect that the use of coresets could lead similar gains for other Bayesian models. Designing coreset algorithms for other widely-used models is an exciting direction for future research.

## Acknowledgments

All authors are supported by the Office of Naval Research under ONR MURI grant N000141110688. JHH is supported by a National Defense Science and Engineering Graduate (NDSEG) Fellowship.

## Footnotes

[1] Recall that the tilde notation suppresses logarithmic terms.

[2]Certain pathological datasets allow us to use unbounded $\Theta$, but we do not assume we are given such data.

[3]More details on our implementation are provided in the Supplementary Material. Code to recreate all of our experiments is available at `https://bitbucket.org/jhhuggins/lrcoresets`.

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
