[Supplementary Material]

# Supplementary Material for
# *Coresets for Scalable Bayesian Logistic Regression*

**Jonathan H. Huggins**     **Trevor Campbell**     **Tamara Broderick**
Computer Science and Artificial Intelligence Laboratory, MIT
{jhuggins@, tdjc@, tbroderick@csail.}mit.edu

## A  Marginal Likelihood Approximation

*Proof of Proposition 2.1.* By the assumption that $\mathcal{L}$ and $\tilde{\mathcal{L}}$ are non-positive, the multiplicative error assumption, and Jensen's inequality,

$$\tilde{\mathcal{E}} = \int e^{\tilde{\mathcal{L}}(\theta)} \pi_0(\theta)\, \mathrm{d}\theta \geq \int e^{(1+\varepsilon)\mathcal{L}(\theta)} \pi_0(\theta)\, \mathrm{d}\theta \geq \left( \int e^{\mathcal{L}(\theta)} \pi_0(\theta)\, \mathrm{d}\theta \right)^{1+\varepsilon} = \mathcal{E}^{1+\varepsilon}$$

and

$$\tilde{\mathcal{E}} = \int e^{\tilde{\mathcal{L}}(\theta)} \pi_0(\theta)\, \mathrm{d}\theta \leq \int e^{(1-\varepsilon)\mathcal{L}(\theta)} \pi_0(\theta)\, \mathrm{d}\theta \leq \left( \int e^{\mathcal{L}(\theta)} \pi_0(\theta)\, \mathrm{d}\theta \right)^{1-\varepsilon} = \mathcal{E}^{1-\varepsilon}.$$

□

## B  Main Results

In order to construct coresets for logistic regression, we will use the framework developed by Feldman and Langberg [3] and improved upon by Braverman et al. [2]. For $n \in [N] := \{1, \dots, N\}$, let $f_n : \mathcal{S} \to \mathbb{R}_+$ be a non-negative function from some set $\mathcal{S}$ and let $\bar{f} = \frac{1}{N} \sum_{n=1}^{N} f_n$ be the average of the functions. Define the *sensitivity* of $n \in [N]$ with respect to $\mathcal{S}$ by

$$\sigma_n(\mathcal{S}) := \sup_{s \in \mathcal{S}} \frac{f_n(s)}{\bar{f}(s)},$$

and note that $\sigma_n(\mathcal{S}) \leq N$. Also, for the set $\mathcal{F} := \{f_n \mid n \in [N]\}$, define the dimension $\dim(\mathcal{F})$ of $\mathcal{F}$ to be the minimum integer $d$ such that

$$\forall F \subseteq \mathcal{F}, \ |\{F \cap R \mid R \in \mathbf{ranges}(\mathcal{F})\}| \leq (|F| + 1)^d,$$

where $\mathbf{ranges}(\mathcal{F}) := \{\mathbf{range}(s, a) \mid s \in \mathcal{S}, a \geq 0\}$ and $\mathbf{range}(s, a) := \{f \in \mathcal{F} \mid f(s) \leq a\}$.

We make use of the following improved version of Feldman and Langberg [3, Theorems 4.1 and 4.4].

**Theorem B.1** (Braverman et al. [2])**.** *Fix $\varepsilon > 0$. For $n \in [N]$, let $m_n \in \mathbb{R}_+$ be chosen such that*

$$m_n \geq \sigma_n(\mathcal{S})$$

*and let $\bar{m}_N := \frac{1}{N} \sum_{n=1}^{N} m_n$. There is a universal constant $c$ such that if $\mathcal{C}$ is a sample from $\mathcal{F}$ of size*

$$|\mathcal{C}| \geq \frac{c\, \bar{m}_N}{\varepsilon^2} (\dim(\mathcal{F}) \log \bar{m}_N + \ln(1/\delta)),$$

*such that the probability that each element of $\mathcal{C}$ is selected independently from $\mathcal{F}$ with probability $\frac{m_n}{N \bar{m}_N}$ that $f_n$ is chosen, then with probability at least $1 - \delta$, for all $s \in \mathcal{S}$,*

$$\left| \bar{f}(s) - \frac{\bar{m}_N}{|\mathcal{C}|} \sum_{f \in \mathcal{C}} \frac{f(s)}{m_n} \right| \leq \varepsilon \bar{f}(s).$$

The set $\mathcal{C}$ in the theorem is called a *coreset*. In our application to logistic regression, $\mathcal{S} = \Theta$ and $f_n(\theta) = -\ln p(Y_n \mid X_n, \theta)$. The key is to determine $\dim(\mathcal{F})$ and to construct the values $m_n$ efficiently. Furthermore, it is necessary for $\bar{m}_N = o(\sqrt{N})$ at a minimum and preferable for $\bar{m}_N = O(1)$.

Letting $Z_n = Y_n X_n$ and $\phi(s) = \ln(1 + \exp(-s))$, we can rewrite $f_n(\theta) = \phi(Z_n \cdot \theta)$. Hence, the goal is to find an upper bound

$$m_n \geq \sigma_n(\Theta) = \sup_{\theta \in \Theta} \frac{N\,\phi(Z_n \cdot \theta)}{\sum_{n'=1}^N \phi(Z_{n'} \cdot \theta)}.$$

To obtain an upper bound on the sensitivity, we will take $\Theta = \mathbb{B}_R$ for some $R > 0$.

**Lemma B.2.** *For all $a, b \in \mathbb{R}$, $\phi(a)/\phi(b) \leq e^{|a-b|}$.*

*Proof.* The lemma is trivial when $a = b$. Let $\Delta = b - a \neq 0$ and $\rho(a) = \phi(a)/\phi(a + \Delta)$. We have

$$\rho'(a) = \frac{(1 + e^a)\log(1 + e^{-a}) - (1 + e^{a+\Delta})\log(1 + e^{-a-\Delta})}{(1 + e^a)(1 + e^{a+\Delta})\log^2(1 + e^{-a-\Delta})}.$$

Examining the previous display we see that $\mathrm{sgn}(\rho'(a)) = \mathrm{sgn}(\Delta)$. Hence if $\Delta > 0$,

$$\begin{aligned}
\sup_a \frac{\phi(a)}{\phi(a + \Delta)} &= \lim_{a \to \infty} \frac{\phi(a)}{\phi(a + \Delta)} \\
&= \lim_{a \to \infty} \frac{\phi'(a)}{\phi'(a + \Delta)} \\
&= \lim_{a \to \infty} \frac{e^{-a}}{1 + e^{-a}} \frac{1 + e^{-a-\Delta}}{e^{-a-\Delta}} \\
&= e^{\Delta} = e^{|b-a|},
\end{aligned}$$

where the second equality follows from L'Hospital's rule. Similarly, if $\Delta < 0$,

$$\begin{aligned}
\sup_a \frac{\phi(a)}{\phi(a + \Delta)} &= \lim_{a \to -\infty} \frac{e^{-a}}{1 + e^{-a}} \frac{1 + e^{-a-\Delta}}{e^{-a-\Delta}} \\
&= \lim_{a \to -\infty} e^{\Delta} \frac{e^{-a}}{e^{-a-\Delta}} \\
&= 1 \leq e^{|b-a|},
\end{aligned}$$

where in this case we have used L'Hospital's rule twice. $\qquad\square$

**Lemma B.3.** *The function $\phi(s)$ is convex.*

*Proof.* A straightforward calculation shows that $\phi''(s) = \frac{e^s}{(1+e^s)^2} > 0$. $\qquad\square$

**Lemma B.4.** *For a random vector $Z \in \mathbb{R}^D$ with finite mean $\bar{Z} = \mathbb{E}[Z]$ and a fixed vectors $V, \theta^* \in \mathbb{R}^D$,*

$$\inf_{\theta \in \mathbb{B}_R} \mathbb{E}\left[\frac{\phi(Z \cdot (\theta + \theta^*))}{\phi(V \cdot (\theta + \theta^*))}\right] \geq e^{-R\|\bar{Z} - V\|_2 - |(\bar{Z} - V) \cdot \theta^*|}.$$

*Proof.* Using Lemmas B.2 and B.3, Jensen's inequality, and the triangle inequality, we have

$$\begin{aligned}
\inf_{\theta \in \mathbb{B}_R} \mathbb{E}\left[\frac{\phi(Z \cdot (\theta + \theta^*))}{\phi(V \cdot (\theta + \theta^*))}\right] &\geq \inf_{\theta \in \mathbb{B}_R} \frac{\phi(\mathbb{E}[Z] \cdot (\theta + \theta^*))}{\phi(V \cdot (\theta + \theta^*))} \\
&\geq \inf_{\theta \in \mathbb{B}_R} e^{-|(\bar{Z} - V) \cdot (\theta + \theta^*)|} \\
&\geq \inf_{\theta \in \mathbb{B}_R} e^{-|(\bar{Z} - V) \cdot \theta| - |(\bar{Z} - V) \cdot \theta^*|} \\
&= e^{-R\|\bar{Z} - V\|_2 - |(\bar{Z} - V) \cdot \theta^*|}.
\end{aligned}$$

$\qquad\square$

We now prove the following generalization of Lemma 3.1

**Lemma B.5.** *For any $k$-clustering $\mathcal{Q}$, $\theta^* \in \mathbb{R}^d$, and $R > 0$,*

$$\sigma_n(\theta^* + \mathbb{B}_R) \le m_n := \left\lceil \frac{N}{1 + \sum_{i=1}^{k} |G_i^{(-n)}| e^{-R\|\bar{Z}_{G,i}^{(-n)} - Z_n\|_2 - |(\bar{Z}_{G,i}^{(-n)} - Z_n)\cdot\theta^*|}} \right\rceil.$$

*Furthermore, $m_n$ can be calculated in $O(k)$ time.*

*Proof.* Straightforward manipulations followed by an application of Lemma B.4 yield

$$\sigma_n(\theta^* + \mathbb{B}_R)^{-1} = \inf_{\theta \in \mathbb{B}_R} \frac{1}{N} \sum_{n'=1}^{N} \frac{\phi(Z_{n'} \cdot (\theta + \theta^*))}{\phi(Z_n \cdot (\theta + \theta^*))}$$

$$= \inf_{\theta \in \mathbb{B}_R} \frac{1}{N} \left[ 1 + \sum_{i=1}^{k} \sum_{Z' \in G_i^{(-n)}} \frac{\phi(Z' \cdot (\theta + \theta^*))}{\phi(Z_n \cdot (\theta + \theta^*))} \right]$$

$$= \inf_{\theta \in \mathbb{B}_R} \frac{1}{N} \left[ 1 + \sum_{i=1}^{k} |G_i^{(-n)}| \, \mathbb{E}\left[ \frac{\phi(Z_{G,i}^{(-n)} \cdot (\theta + \theta^*))}{\phi(Z_n \cdot (\theta + \theta^*))} \right] \right]$$

$$\ge \frac{1}{N} \left[ 1 + \sum_{i=1}^{k} |G_i^{(-n)}| e^{-R\|\bar{Z}_{G,i}^{(-n)} - Z_n\|_2 - |(\bar{Z}_{G,i}^{(-n)} - Z_n)\cdot\theta^*|} \right].$$

To see that the bound can be calculated in $O(k)$ time, first note that the cluster $i_n$ to which $Z_n$ belongs can be found in $O(k)$ time while $\bar{Z}_{G,i_n}^{(-n)}$ can be calculated in $O(1)$ time. For $i \ne i_n$, $G_i^{(-n)} = G_i$, so $\bar{Z}_{G,i}^{(-n)}$ is just the mean of cluster $i$, and no extra computation is required. Finally, computing the sum takes $O(k)$ time. □

In order to obtain an algorithm for generating coresets for logistic regression, we require a bound on the dimension of the range space constructed from the examples and logistic regression likelihood.

**Proposition B.6.** *The set of functions $\mathcal{F} = \{f_n(\theta) = \phi(Z_n \cdot \theta) \mid n \in [N]\}$ satisfies $\dim(\mathcal{F}) \le D+1$.*

*Proof.* For all $F \subseteq \mathcal{F}$,

$$|\{F \cap R \mid R \in \mathbf{ranges}(\mathcal{F})\}| = |\{\mathbf{range}(F, \theta, a) \mid \theta \in \Theta, a \ge 0\}|,$$

where $\mathbf{range}(F, \theta, a) := \{f_n \in \mathcal{F} \mid f_n(\theta) \le a\}$. But, since $\phi$ is invertible and monotonic,

$$\{f_n \in \mathcal{F} \mid f_n(\theta) \le a\} = \{f_n \in \mathcal{F} \mid \phi(Z_n \cdot \theta) \le a\}$$
$$= \{f_n \in \mathcal{F} \mid Z_n \cdot \theta \le \phi^{-1}(a)\},$$

which is exactly a set of points shattered by the hyperplane classifier $Z \mapsto \mathrm{sgn}(Z \cdot \theta - b)$, with $b := \phi^{-1}(a)$. Since the VC dimension of the hyperplane concept class is $D + 1$, it follows that [5, Lemmas 3.1 and 3.2]

$$|\{\mathbf{range}(F, \theta, a) \mid \theta \in \Theta, a \ge 0\}| \le \sum_{j=0}^{D+1} \binom{|F|}{j} \le \sum_{j=0}^{D+1} \frac{|F|^j}{j!}$$

$$\le \sum_{j=0}^{D+1} \binom{D+1}{j} |F|^j = (|F| + 1)^{D+1}.$$

□

*Proof of Theorem 3.2.* Combine Theorem B.1, Lemma 3.1, and Proposition B.6. The algorithm has overall complexity $O(Nk)$ since it requires $O(Nk)$ time to calculate the sensitivities by Lemma 3.1 and $O(N)$ time to sample the coreset. □

## C  Sensitivity Lower Bounds

**Lemma C.1.** *Let $V_1, \ldots, V_K \in \mathbb{R}^{D-1}$ be unit vectors such that for some $\epsilon > 0$, for all $k \neq k'$, $V_k \cdot V_{k'} \leq 1 - \epsilon$. Then for $0 < \delta < \sqrt{1/2}$, there exist unit vectors $Z_1, \ldots, Z_K \in \mathbb{R}^D$ such that*

- *for $k \neq k'$, $Z_k \cdot Z_{k'} \geq 1 - 2\delta^2 > 0$*

- *for $k = 1, \ldots, K$ and $\alpha > 0$, there exists $\theta_k \in \mathbb{R}^D$ such that $\|\theta\|_2 \leq \sqrt{2}\delta\alpha$, $\theta_k \cdot Z_k = -\frac{\alpha\epsilon\delta^2}{2}$ and for $k \neq k$, $\theta_k \cdot Z_{k'} \geq \frac{\alpha\epsilon\delta^2}{2}$.*

*Proof.* Let $Z_k$ be defined such that $Z_{ki} = \delta V_{ki}$ for $i = 1, \ldots, D-1$ and $Z_{kD} = \sqrt{1-\delta^2}$. Thus, $\|Z_k\|_2 = 1$ and for $k \neq k'$,

$$Z_k \cdot Z_{k'} = \delta^2 V_k \cdot V_{k'} + 1 - \delta^2 \geq 1 - 2\delta^2$$

since $V_k \cdot V_{k'} \geq -1$. Let $\theta_k$ be such that $\theta_{ki} = -\alpha\delta V_{ki}$ for $i = 1, \ldots, D-1$ and $\theta_{kd} = \frac{\alpha\delta^2(1-\epsilon/2)}{\sqrt{1-\delta^2}}$. Hence,

$$\theta_k \cdot \theta_k = \alpha^2\delta^2 \left( V_k \cdot V_k + \frac{(1-\epsilon/2)^2\delta^2}{1-\delta^2} \right) \leq 2\alpha^2\delta^2$$

$$\theta_k \cdot Z_k = \alpha(-\delta^2 V_k \cdot V_k + \delta^2(1-\epsilon/2)) = -\frac{\alpha\epsilon\delta^2}{2},$$

and for $k' \neq k$,

$$\theta_k \cdot Z_{k'} = \alpha(-\delta^2 V_k \cdot V_{k'} + \delta^2(1-\epsilon/2)) \geq \alpha\delta^2(-1+\epsilon+1-\epsilon/2) = \frac{\alpha\epsilon\delta^2}{2}.$$

$\square$

**Proposition C.2.** *Let $V_1, \ldots, V_K \in \mathbb{R}^{D-1}$ be unit vectors such that for some $\epsilon > 0$, for all $k \neq k'$, $V_k \cdot V_{k'} \leq 1 - \epsilon$. Then for any $0 < \epsilon' < 1$, there exist unit vectors $Z_1, \ldots, Z_K \in \mathbb{R}^D$ such that for $k, k'$, $Z_k \cdot Z_{k'} \geq 1 - \epsilon'$ but for any $R > 0$,*

$$\sigma_k(\mathbb{B}_R) \geq \frac{K}{1 + (K-1)e^{-R\epsilon\sqrt{\epsilon'}/4}},$$

*and hence $\sigma_k(\mathbb{R}^D) = K$.*

*Proof.* Let $Z_1, \ldots, Z_K \in \mathbb{R}^D$ be as in Lemma C.1 with $\delta$ such that $\delta^2 = \epsilon'/2$. Since for $s \geq 0$, $\phi(s)/\phi(-s) \leq e^{-s}$, conclude that, choosing $\alpha$ such that $\sqrt{2}\alpha\delta = R$, we have

$$
\begin{aligned}
\sigma_n(\mathbb{B}_R) &= \sup_{\theta \in \mathbb{B}_R} \frac{K\,\phi(Z_k \cdot \theta)}{\sum_{k'=1}^{K} \phi(Z_{k'} \cdot \theta)} \\
&\geq \frac{K\,\phi(-\alpha\epsilon\delta^2/2)}{\phi(-\alpha\epsilon\delta^2/2) + (K-1)\phi(\alpha\epsilon\delta^2/2)} \\
&\geq \frac{K}{1 + (K-1)e^{-\alpha\epsilon\delta^2/2}} \\
&= \frac{K}{1 + (K-1)e^{-R\epsilon\sqrt{\epsilon'}/4}}.
\end{aligned}
$$

$\square$

*Proof of Theorem 3.4.* Choose $V_1, \ldots, V_N \in \mathbb{R}^{D-1}$ to be any $N$ distinct unit vectors. Apply Proposition C.2 with $K = N$ and $\epsilon = 1 - \max_{n \neq n'} V_n \cdot V_{n'} > 0$. $\square$

*Proof of Proposition 3.5.* First note that if $V$ is uniformly distributed on $\mathbb{S}^D$, then the distribution of $V \cdot V'$ does not depend on the distribution of $V'$ since $V \cdot V'$ and $V \cdot V''$ are equal in distribution for all $V', V'' \in \mathbb{S}^D$. Thus it suffices to take $V'_1 = 1$ and $V'_i = 0$ for all $i = 2, \ldots, D$. Hence the

distribution of $V \cdot V'$ is equal to the distribution of $V_1$. The CDF of $V_1$ is easily seen to be proportional to the surface area (SA) of $C_s := \{v \in \mathbb{S}^D \mid v_1 \leq s\}$. That is, $\mathbb{P}[V_1 \leq s] = \mathrm{SA}(C_s)/\mathrm{SA}(C_1)$. Let $U \sim \mathsf{Beta}(\frac{D-1}{2}, \frac{1}{2})$, and let $B(a, b)$ be the beta function. It follows from [6, Eq. 1], that by setting $s = 1 - \epsilon$ with $\epsilon \in [0, 1/2]$,

$$
\begin{aligned}
\mathbb{P}[V_1 \geq 1 - \epsilon] &= \frac{1}{2}\mathbb{P}[-\sqrt{1-U} \leq \epsilon - 1] \\
&= \frac{1}{2}\mathbb{P}[U \leq 2\epsilon - \epsilon^2] \\
&= \frac{1}{2B(\frac{D-1}{2}, \frac{1}{2})} \int_0^{2\epsilon-\epsilon^2} t^{(D-3)/2}(1-t)^{-1/2}\, \mathrm{d}t \\
&\leq \frac{1}{2B(\frac{D-1}{2}, \frac{1}{2})}(1-\epsilon)^{-1} \int_0^{2\epsilon-\epsilon^2} t^{(D-3)/2}\, \mathrm{d}t \\
&= \frac{1}{(D-1)B(\frac{D-1}{2}, \frac{1}{2})} \frac{(2-\epsilon)^{(D-1)/2}}{1-\epsilon}\epsilon^{(D-1)/2} \\
&\leq \frac{2^{(D+1)/2}}{(D-1)B(\frac{D-1}{2}, \frac{1}{2})}\epsilon^{(D-1)/2}.
\end{aligned}
$$

Applying a union bound over the $\binom{D}{2}$ distinct vector pairs completes the proof. $\square$

**Lemma C.3** (Hoeffding's inequality [1, Theorem 2.8]). *Let $A_k$ be zero-mean, independent random variables with $A_k \in [-a, a]$. Then for any $t > 0$,*

$$
\mathbb{P}\left(\sum_{k=1}^{K} A_k \geq t\right) \leq e^{-\frac{t^2}{2a^2 K}}.
$$

*Proof of Proposition 3.6.* We say that unit vectors $V$ and $V'$ are $(1-\epsilon)$-*orthogonal* if $|V \cdot V'| \leq 1 - \epsilon$. Clearly $\|V_n\|_2 = 1$. For $n \neq n'$, by Hoeffding's inequality $\mathbb{P}(|V_n \cdot V_{n'}| \geq 1 - \epsilon) \leq 2e^{-(1-\epsilon)^2 D/2}$. Applying a union bound to all $\binom{K}{2}$ pairs of vectors, the probability that any pair is not $(1-\epsilon)$-orthogonal is at most

$$
2\binom{K}{2}e^{-(1-\epsilon)^2 D/2} \leq \frac{1}{2}.
$$

Thus, with probability at least $1/2$, $V_1, \ldots, V_N$ are pairwise $(1-\epsilon)$-orthogonal. $\square$

*Proof of Corollary 3.7.* The data from Theorem 3.4 satisfies $Z_n \cdot Z_{n'} \geq 1 - \epsilon'$, so for $n \neq n'$,

$$
\|Z_n - Z_{n'}\|_2^2 = 2 - 2Z_n \cdot Z_{n'} \leq 2\epsilon'.
$$

Applying Lemma 3.1 with the clustering $\mathcal{Q} = \{Z_1, \ldots, Z_N\}$ and combining it with the lower bound in Theorem 3.4 yields the result. $\square$

# D   A Priori Expected Sensitivity Upper Bounds

*Proof of Proposition 3.8.* First, fix the number of datapoints $N \in \mathbb{N}$. Since $X_n$ are generated from a mixture, let $L_n$ denote the integer mixture component from which $X_n$ was generated, let $C_i$ be the set of integers $1 \leq j \leq N$ with $j \neq n$ and $L_j = i$, and let $C = (C_i)_{i=1}^{\infty}$. Note that with this definition, $|G_i^{(-n)}| = |C_i|$. Using Jensen's inequality and the upper bound from Lemma 3.1 with

the clustering induced by the label sequence,

$$\mathbb{E}\left[\sigma_n\left(\mathbb{B}_R\right)\right] \leq \mathbb{E}\left[m_n\right] = N\mathbb{E}\left[\frac{1}{1 + \sum_i |C_i| e^{-R\|\bar{Z}_{G,i}^{(-n)} - Z_n\|_2}}\right]$$

$$= N\mathbb{E}\left[\mathbb{E}\left[\frac{1}{1 + \sum_i |C_i| e^{-R\|\bar{Z}_{G,i}^{(-n)} - Z_n\|_2}} \mid C\right]\right]$$

$$\leq N\mathbb{E}\left[\frac{1}{1 + \sum_i |C_i| e^{-R\mathbb{E}\left[\|\bar{Z}_{G,i}^{(-n)} - Z_n\|_2 \mid C\right]}}\right].$$

Using Jensen's inequality again and conditioning on the labels $Y = (Y_n)_{n=1}^N$ and indicator $L_n$,

$$\mathbb{E}\left[\|\bar{Z}_{G,i}^{(-n)} - Z_n\|_2 \mid C\right] \leq \sqrt{\mathbb{E}\left[\|\bar{Z}_{G,i}^{(-n)} - Z_n\|_2^2 \mid C\right]}$$

$$= \sqrt{\mathbb{E}\left[\mathbb{E}\left[\|\bar{Z}_{G,i}^{(-n)} - Z_n\|_2^2 \mid C, L_n, Y\right] \mid C\right]}.$$

For fixed labels $Y$ and clustering $C$, $L_n$, the linear combination in the expectation is multivariate normal with

$$\bar{Z}_{G,i}^{(-n)} - Z_n \sim \mathcal{N}\left(\frac{1}{|C_i|}\left(\sum_{j \in C_i} Y_j\right)\mu_i - Y_n\mu_n', \frac{1}{|C_i|}\Sigma_i + \Sigma_n'\right),$$

where $\mu_n', \Sigma_n'$ are the mean and covariance of the mixture component that generated $X_n$. Further, for any multivariate normal random vector $W \in \mathbb{R}^d$,

$$\mathbb{E}\left[W^T W\right] = \sum_{m=1}^d \mathbb{E}\left[W_m^2\right] = \sum_{m=1}^d \text{Var}\left[W_m\right] + \mathbb{E}\left[W_m\right]^2,$$

so

$$\mathbb{E}\left[\|\bar{Z}_{G,i}^{(-n)} - Z_n\|_2^2 \mid L_n, C, Y\right]$$

$$= \text{Tr}\left[\frac{1}{|C_i|}\Sigma_i + \Sigma_n'\right] + \left(\frac{\sum_{j \in C_i} Y_j}{|C_i|}\right)^2 \mu_i^T \mu_i - 2Y_n\left(\frac{\sum_{j \in C_i} Y_j}{|C_i|}\right)\mu_i^T \mu_n' + \mu_n'^T \mu_n'.$$

Exploiting the i.i.d.-ness of $Y_j$ for $j \in C_i$ given $C$, defining $\bar{y}_j = \mathbb{E}\left[Y_i | L_i = j\right]$, and noting that $X_n$ is sampled from the mixture model,

$$\mathbb{E}\left[\mathbb{E}\left[\|\bar{Z}_{G,i}^{(-n)} - Z_n\|_2^2 \mid L_n, C, Y\right] \mid C\right]$$

$$= \sum_j \pi_j \left(\text{Tr}\left[\frac{1}{|C_i|}\Sigma_i + \Sigma_j\right] + \frac{|C_i|\bar{y}_i^2 + 1 - \bar{y}_i^2}{|C_i|}\mu_i^T \mu_i - 2\bar{y}_j\bar{y}_i\mu_i^T \mu_j + \mu_j^T \mu_j\right)$$

$$= \sum_j \pi_j \left(\frac{\text{Tr}\left[\Sigma_i\right] + \left(1 - \bar{y}_i^2\right)\mu_i^T \mu_i}{|C_i|} + \text{Tr}\left[\Sigma_j\right] + \bar{y}_i^2\mu_i^T \mu_i - 2\bar{y}_j\bar{y}_i\mu_i^T \mu_j + \mu_j^T \mu_j\right)$$

$$= A_i |C_i|^{-1} + B_{in},$$

where $A_i$ and $B_i$ are positive constants

$$A_i = \text{Tr}\left[\Sigma_i\right] + \left(1 - \bar{y}_i^2\right)\mu_i^T \mu_i$$

$$B_i = \sum_j \pi_j \left(\text{Tr}\left[\Sigma_j\right] + \bar{y}_i^2\mu_i^T \mu_i - 2\bar{y}_i\bar{y}_j\mu_i^T \mu_j + \mu_j^T \mu_j\right).$$

Therefore, with $0^{-1}$ defined to be $+\infty$,

$$\mathbb{E}\left[m_n\right] \leq N\mathbb{E}\left[\frac{1}{1 + \sum_i |C_i| e^{-R\sqrt{A_i|C_i|^{-1} + B_i}}}\right].$$

As $N \to \infty$, we expect the values of $|C_i|/N$ to concentrate around $\pi_i$. To get a finite sample bound using this intuition, we split the expectation into two conditional expectations: one where all $|C_i|/N$ are not too far from $\pi_i$, and one where they may be. Define $g : \mathbb{R}_+^\infty \to \mathbb{R}_+$ as

$$g(x) = \frac{1}{1 + \sum_i x_i e^{-R\sqrt{A_i x_i^{-1} + B_i}}},$$

$\pi = (\pi_1, \pi_2, \dots)$, $\epsilon = (\epsilon_1, \epsilon_2, \dots)$ with $\epsilon_i > 0$, and $\eta_i = \max(\pi_i - \epsilon_i, 0)$. Then

$$\mathbb{E}\left[m_n\right] \leq N\mathbb{P}\left(\forall i, \frac{|C_i|}{N} \geq \eta_i\right) g(N\eta) + N\mathbb{P}\left(\exists i : \frac{|C_i|}{N} < \eta_i\right)$$

$$= Ng(N\eta) + N\mathbb{P}\left(\exists i : \frac{|C_i|}{N} < \eta_i\right)(1 - g(N\eta)).$$

Using the union bound, noting that $1 - g(N\eta) \leq 1$, and then using Hoeffding's inequality yields

$$\mathbb{E}\left[m_n\right] \leq Ng(N\eta) + N\sum_i \mathbb{P}\left(\frac{|C_i|}{N} < \eta_i\right)$$

$$\leq Ng(N\eta) + N\sum_{i:\pi_i > \epsilon_i} \mathbb{P}\left(\frac{|C_i|}{N} - \pi_i < -\epsilon_i\right)$$

$$\leq Ng(N\eta) + N\sum_{i:\pi_i > \epsilon_i} e^{-2N\epsilon_i^2}$$

$$= \frac{1}{N^{-1} + \sum_i \eta_i e^{-R\sqrt{A_i N^{-1}\eta_i^{-1} + B_i}}} + \sum_{i:\pi_i > \epsilon_i} Ne^{-2N\epsilon_i^2}.$$

We are free to pick $\epsilon$ as a function of $\pi$ and $N$. Let $\epsilon = N^{-r}$ for any $0 < r < 1/2$. Note that this means $\eta_i = \max(\pi_i - N^{-r}, 0)$. Then

$$\mathbb{E}\left[m_n\right] = \frac{1}{N^{-1} + \sum_i \eta_i e^{-R\sqrt{A_i N^{-1}\eta_i^{-1} + B_i}}} + \sum_{i:\eta_i > 0} Ne^{-2N^{1-2r}}.$$

It is easy to see that the first term converges to $\left(\sum_i \pi_i e^{-R\sqrt{B_i}}\right)^{-1}$ by a simple asymptotic analysis. To show the second term converges to 0, note that for all $N$,

$$\sum_i \pi_i = \sum_{i:\pi_i > N^{-r}} \pi_i + \sum_{i:\pi_i \leq N^{-r}} \pi_i$$

$$\geq \sum_{i:\pi_i > N^{-r}} \pi_i$$

$$\geq \sum_{i:\pi_i > N^{-r}} N^{-r}$$

$$= \left|\{i : \pi_i > N^{-r}\}\right| N^{-r}.$$

Since $\sum_i \pi_i = 1 < \infty$, $|\{i : \pi_i > N^{-r}\}| = O(N^r)$. Therefore there exists constants $C, M < \infty$ such that

$$\left|\{i : \pi_i > N^{-r}\}\right| \leq M + CN^r,$$

and thus

$$\sum_{i:\pi_i > N^{-r}} Ne^{-2N^{1-2r}} \leq N(M + CN^r)e^{-2N^{1-2r}} \to 0, \qquad N \to \infty.$$

Finally, since $\bar{m}_N = \frac{1}{N}\sum_{n=1}^N m_n$, we have $\mathbb{E}\left[\bar{m}_N\right] = \mathbb{E}\left[m_n\right]$, and the result follows. $\qquad\square$

*Proof of Corollary 3.9.* This is a direct result of Proposition 3.8 with $\pi_1 = 1$, $\pi_i = 0$ for $i \geq 2$. $\quad\square$

Table 1: Datasets used for experiments

| Name | $N$ | $D$ | positive examples | $k$ |
|---|---|---|---|---|
| Low-dimensional Synthetic Binary | 1M | 5 | 9.5% | 4 |
| Higher-dimensional Synthetic Binary | 1M | 10 | 8.9% | 4 |
| Synthetic Balanced Mixture | 1M | 10 | 50% | 4 |
| Chemical Reactivity[2] | 26,733 | 100 | 3% | 6 |
| Webspam[3] | 350K | 127 | 60% | 6 |
| Cover type[4] | 581,012 | 54 | 51% | 6 |

# E   Further Experimental Details

The datasets we used are summarized in Table 1. We briefly discuss some implementation details of our experiments.

**Implementing Algorithm 1.** One time-consuming part of creating the coreset is calculating the adjusted centers $\bar{Z}_{G,i}^{(-n)}$. We instead used the original centers $Q_i$. Since we use small $k$ values and $N$ in large, each cluster is large. Thus, the difference between $\bar{Z}_{G,i}^{(-n)}$ and $Q_i$ was negligible in practice, resulting at most a 1% change in the sensitivity while resulting in an order of magnitude speed-up in the algorithm. In order to speed up the clustering step, we selected a random subset of the data of size $L = \min(1000k, 0.025N)$ and ran the `sklearn` implementation of $k$-means++ to obtain $k$ cluster centers. We then calculated the clustering and the normalized $k$-means score $\mathcal{I}$ for the full dataset. Notice that $L$ is chosen to be independent of $N$ as $N$ becomes large but is never more than a construct fraction of the full dataset when $N$ is small.[1] Thus, calculating a clustering only takes a small amount of time that is comparable to the time required to run our implementation of Algorithm 1.

**Posterior Inference Procedure.** We used the adaptive Metropolis-adjusted Langevin algorithm [4, 8], where we adapted the overall step size and targeted an acceptance rate of 0.574 [7]. It $T$ iterations were used in total, adaptation was done for the first $T/2$ iterations while the remaining iterations were used as approximate posterior samples. For the subsampling experiments, for a subsample size $M$, an approximate dataset $\tilde{\mathcal{D}}$ of size $M$ was obtained either using random sampling or Algorithm 1. The dataset $\tilde{\mathcal{D}}$ was then fixed for the full MCMC run.

## Footnotes

[1]Note that we use data subsampling here *only* to choose the cluster centers. We still calculate sensitivity upper bounds across the entire data set and thereby are still able to capture rare but influential data patterns. Indeed, we expect influential data points to be far from cluster centers chosen either with or without subsampling, and we thereby expect to pick up these data points with high probability during the coreset sampling procedure in Algorithm 1.

[2]Dataset `ds1.100` from `http://komarix.org/ac/ds/`.

[3]Available from `http://www.cc.gatech.edu/projects/doi/WebbSpamCorpus.html`

[4]Dataset `covtype.binary` from `https://www.csie.ntu.edu.tw/~cjlin/libsvmtools/datasets/binary.html`.

[7] G. O. Roberts and J. S. Rosenthal. Optimal scaling for various Metropolis-Hastings algorithms. *Statistical Science*, 16(4):351–367, 2001.

[8] G. O. Roberts and R. L. Tweedie. Exponential convergence of Langevin distributions and their discrete approximations. *Bernoulli*, 2(4):341–363, Nov. 1996.