[Reviews · NeurIPS 2016]

Reviewer 1

Summary

The paper proposes a definition of a coreset in the context of Bayesian inference, and introduces an algorithm for constructing such coresets for Bayesian logistic regression, which is analyzed theoretically and studied empirically. The analysis revolves around upper and lower bounds for a sensitivity quantity that controls the size of the coreset. Intuitively, the lower bound results show that in order to construct non-trivial coresets the parameter space for logistic regression needs to be bounded, e.g. to within a ball of a certain radius.

Qualitative Assessment

Although simple, the proposed definition of a coreset in the context of Bayesian inference, and analysis of the fidelity to the true posterior, are foundational contributions which open up an exciting and potentially important new line of research, speeding up Bayesian inference using coresets. The remainder of the paper focuses on Bayesian logistic regression, which can be understood as a simple yet important use case for the general theory introduced in Section 2. The sensitivity analysis, with upper and lower bounds, is insightful as well as leading to a practical algorithm. The experimental results are perhaps the weakest portion of the paper. The results would be more compelling if more data sets were used. Right now, I cannot tell if the data sets were cherry-picked. The method performed well on one data set and did not beat (or lose to) naive random sampling on the other data set. While these results suffice to show that the method can be useful in practice, they do not do much to provide insight on which scenarios the technique should be expected to perform well or poorly on.

Confidence in this Review

2-Confident (read it all; understood it all reasonably well)


Reviewer 2

Summary

It is well known that Bayesian methods have intractability issues with regards to scaling. The authors break down the existing data down into its redundant components, known as coresets. The authors propose a coreset construction algorithm for logistic regression, giving theoretical guarantees and showing some success on known datasets and simulation studies.

Qualitative Assessment

This paper is very promising, however, we have some comments and questions regarding the paper. 1. First, we thank the authors for providing overall what is a very clear and articulate paper. Most of our questions deal with the application of the proposed algorithm, we we will focus directly on this. 2. In section 4, line 219, the authors consider simulation studies. a. The explanation of the simulation study and parameters that we examined is not currently clear. How sensitive is your model versus the random model for the values that you choose? How were these values chosen? This section needs much unpacking and more explanation. b. Could the authors please comment on how the model perform under model missspecification compared to a random algorithm? In essence, how robust is the model? c. In general, we believe that there is too little information put on the data sets themselves. There is _very little information about each data set and a description. As a general comment, what are the major limitations of each of the datasets currently and why is your method a great improvement? d. Furthermore, does your method show to be a great improvement on a large dataset (in the millions or billions) as prefaced in the introduction, and can you illustrate the success of this? e. In terms of all your methods, what is the exactly scalability of your method in terms of run time compared to the randomized method (and not just computational complexity). Users would want to know this for practical usability. f. Finally, what would be most useful also to know are the major strengths and weakness of your methods and the practical usability of the method in practice. g. How can the method be extended to other approached beyond logistic regression? What about probit? Is there a general framework that can be developed here that would make the paper much stronger and much more desirable for the larger community to use?

Confidence in this Review

3-Expert (read the paper in detail, know the area, quite certain of my opinion)


Reviewer 3

Summary

The authors propose an algorithm for constructing coresets for Bayesian logistic regression. A coreset is a weighted subset of a dataset such that the weighted log-likelihood computed using this subset is approximately equal to the log-likelihood of the complete dataset. The datapoints in the coreset are chosen based on a sensitivity measure which is inversely related to the redundancy of the datapoint. Since the sensitivity is computationally intractable, the proposed algorithm relies on an upper bound computed using a clustering of the datapoints. The (normalized) upper-bounds are then used as sampling probabilities to create the coreset. Various theoretical results such as the tightness of the upper-bound and a bound on the coreset size are proved. Experiments show better performance compared to a random baseline.

Qualitative Assessment

Coresets have been studied in an optimization context before, but haven't been used by the Bayesian sampling/variational inference communities. The paper is reasonably clear and well written. This line of research is obviously impactful to Bayesian inference, as many MCMC / variational inference algorithms that use the whole dataset can now make updates using only a coreset and be considerably faster. The approach is specific to logistic regression, nevertheless the paper may inspire work on coreset algorithms for other models. The experiments only compare to a random baseline, which is a little disappointing. Comments/Questions: - Gradient of the log-likelihood is more interesting than the log-likelihood in many cases, e.g. gradient based MCMC. Would building a coreset based on approximating the gradient of the log likelihood work better than approximating the log likelihood? - In the MALA experiments, was the corset used both for the gradient step and the Metropolis-Hastings test? - In the MALA experiments, for the random baseline: Is a different random batch chosen in each iteration like in stochastic gradient methods? Or do you use a fixed subset to be more comparable to coresets? - I think you should compare to a smarter baseline than random sampling. How about comparing to a heuristic such as a subset created by picking a few representative elements from different clusters? Or, you mention that coreset like ideas have been previously explored for logistic regression in reference [15]. Can this act as a baseline (I haven't read [15])? - Do you have any theories on why the coreset algorithm is comparable in performance to random sampling on the Webspam dataset, but does much better on Reactivity? Are there any characteristics of the dataset that help determine if there is an advantage to using your algorithm e.g. sparse data, outliers?

Confidence in this Review

2-Confident (read it all; understood it all reasonably well)


Reviewer 4

Summary

This paper proposes a scalable algorithm to construct coresets for Bayesian inference of logistic regression models. This algorithm assumes that the parameters of the model are bounded by a given constant. Likelihood functions of coresets are used as approximations of likelihood functions of actual datasets in MCMC. Besides, conditions for the coreset to be small are studied. The proposed methods are compared with random subsampling MCMC.

Qualitative Assessment

Major problems: 1) Theorem 3.2 only ensures that the resulting likelihood function is probable to be a good approximation. This PAC-type error bound can not guarantee that any resulting coreset is able to well approximate the full dataset. In practice, only one coreset is produced, which may be a bad approximation, and it is used in all steps of MCMC algorithm. So, the inference built upon only one probably correct coreset is not reliable. 2) Theorem 3.2 shows that the proposed algorithm, w.p. at least 1-$\delta$, can produce a $\epsilon$-coreset with size M, which is a function of $\delta$ & $\epsilon$ (line 5 of Alg. 1). In practice, Alg. 1 is used given fixed M without setting values for $\delta$ & $\epsilon$. Could you show, for your examples, that within a preset probability (e.g. 95% or 99%), what value of $\epsilon$ would be for the resulting coreset? or show that within what probability, the resulting coreset would have its $\epsilon$ no greater than a preset value (e.g. 0.1)? It is better to use these quantities to measure the error. 3) For the reactivity data, subsampling MCMC approaches gave unexpected results as their performances deteriorated when the subset size increased. I can not understand why this was happening? Shouldn't their performance increase? Have you checked the convergence of Markov chains for those cases? Similar unexpected result can be found in mixture data example for the case of the subset size be around 100. Besides these unexpected results, only simulated binary data can show the advantage of using coresets, which can not support the claim in the paper that the coreset approach is better. Minor problems: 1) In line 5 of Alg. 1, what is the c? 2) In Figure 1, legends for the first two graphs are missing. 3) In Figure 2, the short vertical lines are not explained. Are they the ranges of the results for 10 different MCMC runs?

Confidence in this Review

2-Confident (read it all; understood it all reasonably well)


Reviewer 5

Summary

This paper aims to develop a method to construct a coreset for Bayesian logistic regression. A coreset is a weighted subset of data, which is hopefully much smaller than the original dataset, and which yields provably similar results when existing inference algorithms are applied to it. Intuitively, this procedure leverages a k-clustering of the data (which must first be constructed) to compute a weight for each data point and to compute the total size of the coreset. The coreset is then resampled from the initial dataset using the computed weights. This paper restricts its focus to the Bayesian logistic regression model only. Furthermore, the theory restricts the parameter space to be within a Euclidean ball of a pre-specified radius R (as opposed to unbounded R^D space of parameters, which is the typical setting for logistic regression). The authors also provide negative theoretical results showing that their theoretical framework would not work on an unbounded parameter space, to justify the bounded parameter assumption. Empirically, the authors provide empirical results on three datasets, which aim to show that inference on coresets performs better than on random subsampling of the data, and provides a good approximation to inference on the full data when the coreset size grows large enough.

Qualitative Assessment

* I have one major gripe with the practicality of using coresets versus recent scalable inference strategies, such as minibatch inference methods. Note that in minibatch inference methods, at each iteration, a small subset of the data is sampled from the full dataset and used to make an update; these methods take advantage of the redundancy in data to perform inexpensive updates. In this paper, coresets reduce the total dataset size by, in some sense, approximating the dataset with a smaller group of (weighted) examples. However, when coresets are used in existing inference algorithms (such as these minibatch algorithms), it seems to me that a very similar procedure will occur: a small subset of this approximate, weighted dataset will be drawn, and used to make an update. I am not convinced this would actually speed up inference (i.e. decrease the number of iterations needed or the time of each iteration), and I don't see a great justification given in this paper. In a sense, I feel that the main thing happening here is that the data is approximated in a smaller/compressed fashion; I can see how this might help with data storage concerns, but I don't see a great justification for why it would appreciably speed inference over existing minibatch methods (especially considering a coreset must be constructed before inference can proceed, which adds additional inference time to this method). One way to demonstrate this would be with timing comparison plots that explicitly show that coresets yield faster inferences given large datasets when compared to minibatch methods---however, no direct experiments of this sort are given. To summarize this complaint, I feel there is some disconnect between what this paper showed and what it set out to do: we know that smaller datasets typically allow for faster inference, and we know that coresets produces smaller (weighted) datasets. However, it’s not convincingly shown that when coresets are used with (or in comparison with) recent efficient methods, they actually allow for faster inference. * After the coreset is constructed, the authors intend for existing inference algorithms to be applied to the coreset. However, it is not immediately clear to me how a weighted set of data can be directly used with existing inference algorithms. One could duplicate or resample data points based on their weight, and use this new dataset directly in existing inference algorithms --- is this what happens? Or is there a general-purpose way to use the coreset weights in existing inference algorithms? * Regarding the definition of an \epsilon-coreset: on page 3, an \epsilon-coreset is defined to be a coreset in which the difference between the log marginal likelihoods of the coreset and the full data are bounded. Intuitively, I feel the most straightforward goal is to design a coreset in which the approximate coreset posterior is similar to the posterior of the full dataset (i.e. the coreset posterior and true posterior are bounded). However, I have no intuition about how bounding the difference between the log marginal likelihoods corresponds to this goal. Perhaps these two bounds do directly correspond in some way; if so, I feel the paper should do more to explicity state how well an \epsilon-coreset posterior approximates the true posterior. Or at least provide more justification about why bounding the log marginal likelihoods is the appropriate goal. * Regarding empirical results: on page 7, the main body of empirical results show that a big enough subset eventually yields a good approximation to the test log-likelihood. However, as I describe above in more detail, the results do not seem to focus on what I feel is the direct goal of this work (which is to allow for faster inference in Bayesian logistic regression by first constructing a coreset). Results of this sort would aim to show that a coreset can be constructed, and inference on this coreset completed, in a faster time than it would take to perform inference on the full original dataset. I feel that results of this sort are particularly important since there exists many popular minibatch algorithms developed in recent times, in which each sample can typically be generated very quickly. It would be nice to see if running these minibatch inference algorithms on a coreset versus the entire dataset has a meaningful speedup. * Experimental comparisons: there were not many experimental comparisons. The main experimental comparison seems to be random subsampling --- it slightly confused me that (I don't believe) this comparison was explicitly described in the paper. However, I assume this is just inference run on a random (uniformly drawn) subset of data of size M. * What is a good way to determine the parameter space radius R (which is required as an input to the coreset algorithm)? The authors mention that restricting the parameter space to a Euclidean ball is justified since the data is usually preprocessed (to be mean zero and variance 1), which causes each parameter component to be small (e.g. absolute value less than 4) --- however, this statement is not rigorously justified, and I am not sure if it is always true. Is there any theoretically justified way to choose a correct R? * It seems like the coreset procedure described in this paper is mainly dependent on how well the data can be approximated by a set of clusters. Theoretically speaking, is there any intuition about a good way to choose the number of clusters, and is the procedure dependent upon this choice? I do, however, notice that the choice of k over a range is demonstrated empirically.

Confidence in this Review

2-Confident (read it all; understood it all reasonably well)


Reviewer 6

Summary

This paper proposes a novel approach to scalable Bayesian inference, in particular Bayesian logistic regression. Assuming that a large part of dataset is redundant, this paper proposes an algorithm to construct a weighted subset of the data (called coreset) that is a) much smaller than the original dataset and b) approximates the log likelihood of the original dataset well. This paper provides theoretical results and experimental results for this algorithm.

Qualitative Assessment

This is a very interesting idea that could prove to be influential. The theoretical analysis is thorough but the paper would benefit from more experimental results. In particular it would be interesting to see this algorithm performs for larger datasets. Why doesn't the algorithm perform well on the webspam dataset, the largest dataset considered?

Confidence in this Review

2-Confident (read it all; understood it all reasonably well)